# Natural silencing of quorum-sensing activity protects *Vibrio parahaemolyticus* from lysis by an autoinducer-detecting phage

Olivia P. Duddy[1], Justin E. Silpe[1,2], Chenyi Fei[1], Bonnie L. Bassler[1,2]*

1 Department of Molecular Biology, Princeton University, Princeton, New Jersey, United States of America,
2 Howard Hughes Medical Institute, Chevy Chase, Maryland, United States of America

* bbassler@princeton.edu

**Data Availability Statement:** The authors confirm that all data underlying the findings are fully available without restriction. All relevant data are within the paper and its Supporting Information files. Data points used to make plots, raw RNA-

## Abstract

Quorum sensing (QS) is a chemical communication process that bacteria use to track population density and orchestrate collective behaviors. QS relies on the production, accumulation, and group-wide detection of extracellular signal molecules called autoinducers. Vibriophage 882 (phage VP882), a bacterial virus, encodes a homolog of the *Vibrio* QS receptor-transcription factor, called VqmA, that monitors the *Vibrio* QS autoinducer DPO. Phage VqmA binds DPO at high host-cell density and activates transcription of the phage gene *qtip*. Qtip, an antirepressor, launches the phage lysis program. Phage-encoded VqmA when bound to DPO also manipulates host QS by activating transcription of the host gene *vqmR*. VqmR is a small RNA that controls downstream QS target genes. Here, we sequence *Vibrio parahaemolyticus* strain O3:K6 882, the strain from which phage VP882 was initially isolated. The chromosomal region normally encoding *vqmR* and *vqmA* harbors a deletion encompassing *vqmR* and a portion of the *vqmA* promoter, inactivating that QS system. We discover that *V. parahaemolyticus* strain O3:K6 882 is also defective in its other QS systems, due to a mutation in *luxO*, encoding the central QS transcriptional regulator LuxO. Both the *vqmR-vqmA* and *luxO* mutations lock *V. parahaemolyticus* strain O3:K6 882 into the low-cell density QS state. Reparation of the QS defects in *V. parahaemolyticus* strain O3:K6 882 promotes activation of phage VP882 lytic gene expression and LuxO is primarily responsible for this effect. Phage VP882-infected QS-competent *V. parahaemolyticus* strain O3:K6 882 cells lyse more rapidly and produce more viral particles than the QS-deficient parent strain. We propose that, in *V. parahaemolyticus* strain O3:K6 882, constitutive maintenance of the low-cell density QS state suppresses the launch of the phage VP882 lytic cascade, thereby protecting the bacterial host from phage-mediated lysis.

## Author summary

Quorum sensing is a bacterial communication process that involves the production, detection, and group-wide response to extracellular signal molecules called autoinducers. Recent studies demonstrate that bacteria-infecting viruses, called phages, encode

sequencing data, and unprocessed gels and blots used in this study are available as Data S1, S2, and S3, respectively.

**Funding:** The authors acknowledge financial support from the Jane Coffin Childs Memorial Fund for Medical Research (J.E.S), Howard Hughes Medical Institute (J.E.S and B.L.B), National Science Foundation grant MCB-2043238 (B.L.B.), the National Institutes of Health grant R37GM065859 (B.L.B.), and NIGMS training grant T32GM007388 (O.P.D.). The funders had no role in study design, data collection and analysis, decision to publish, or preparation of the manuscript.

**Competing interests:** The authors have declared that no competing interests exist.

quorum-sensing receptors that recognize host-produced autoinducers. Detection of host autoinducers drives the phage lysogeny-to-lysis transition. The founding phage demonstrating this quorum-sensing "eavesdropping" mechanism was vibriophage VP882 in its host, *Vibrio parahaemolyticus* strain 882. Here, we show that *V. parahaemolyticus* strain 882 possesses a mechanism that enables it to evade phage "eavesdropping". Unlike other *V. parahaemolyticus* strains, strain 882 harbors multiple mutations in quorum-sensing genes that, together, lock *V. parahaemolyticus* strain 882 into the low cell density quorum-sensing state, preventing it from responding to autoinducers. A consequence is suppression of the phage VP882 lytic program. Indeed, repair of the *V. parahaemolyticus* strain 882 quorum-sensing pathways promotes phage VP882 lytic activity. We propose that the locked low cell density quorum-sensing state of the host silences the information encoded in autoinducers for both *V. parahaemolyticus* strain 882 and for phage VP882, most critically, suppressing the phage lysogeny to lysis transition.

## Introduction

*Vibrio parahaemolyticus* is the major worldwide cause of seafood-borne bacterial gastroenteritis [1]. The first pandemic of this pathogen was caused by *V. parahaemolyticus* serotype O3:K6 [2]. The marine environment, which *V. parahaemolyticus* naturally occupies, abounds with bacterial viruses called phages, and indeed, both lytic and temperate phages have been identified in pandemic *V. parahaemolyticus* O3:K6 strains. Phages are central to bacterial evolution, and their existence is often linked to the emergence of toxigenic bacteria from non-toxigenic strains [3–5]. Phages can benefit their host bacteria by endowing them with traits that promote niche colonization and spread to new territory. Phages can also parasitize their hosts by exploiting resources for their own replication and dissemination to uninfected bacteria. One such phage is the temperate, plasmid-like vibriophage 882 (here forward called phage VP882) isolated from *V. parahaemolyticus* strain O3:K6 882 [6]

Phage VP882 regulates its lysis-lysogeny decision by "eavesdropping" on a host-produced quorum-sensing (QS) autoinducer (AI) molecule called DPO (3,5-dimethylpyrazin-2-ol, produced by the Tdh synthase; Fig 1A and 1B) and [7,8]). Phage VP882 encodes the DPO-binding receptor and transcription factor called VqmA$_{Phage}$ which is a homolog of the bacterial DPO QS receptor VqmA [8]. The bacterial VqmA QS pathway is considered ubiquitous within the *Vibrio* genus, however, to date, it has only been characterized in the species it was discovered, *Vibrio cholerae* [7,9]. *V. cholerae* produces DPO and detects it via VqmA (hereafter called VqmA$_{Vc}$). DPO-bound VqmA$_{Vc}$ activates transcription of *vqmR* (*vqmR$_{Vc}$*), encoding a small regulatory RNA, VqmR$_{Vc}$ (Fig 1A). VqmR$_{Vc}$ represses translation of genes required for the production of virulence factors and formation of biofilms [9,10]. When phage VP882 infects *Vibrio* bacteria, it introduces a second *vqmA* into the system, *vqmA$_{Phage}$*. The current model for the phage-encoded pathway is that an unknown cue induces expression of *vqmA$_{Phage}$* in the infected *Vibrios*. Once produced, VqmA$_{Phage}$ binds host-produced DPO and launches the phage lytic program by activating the expression of an antirepressor-encoding gene called *qtip* [8]. In addition to *qtip*, VqmA$_{Phage}$ also activates expression of host *vqmR$_{Vc}$*, demonstrating that phage VP882 influences both its own and its host's QS pathways (Fig 1A). By contrast, the VqmA$_{Vc}$ protein only binds to its partner *vqmR$_{Vc}$* promoter, not to the phage *qtip* promoter (Fig 1A and [8,11]).

Regarding the phage lysis pathway, Qtip sequesters the cI$_{VP882}$ phage repressor of lysis (Fig 1B). The host SOS response also launches the phage lytic pathway via RecA-mediated

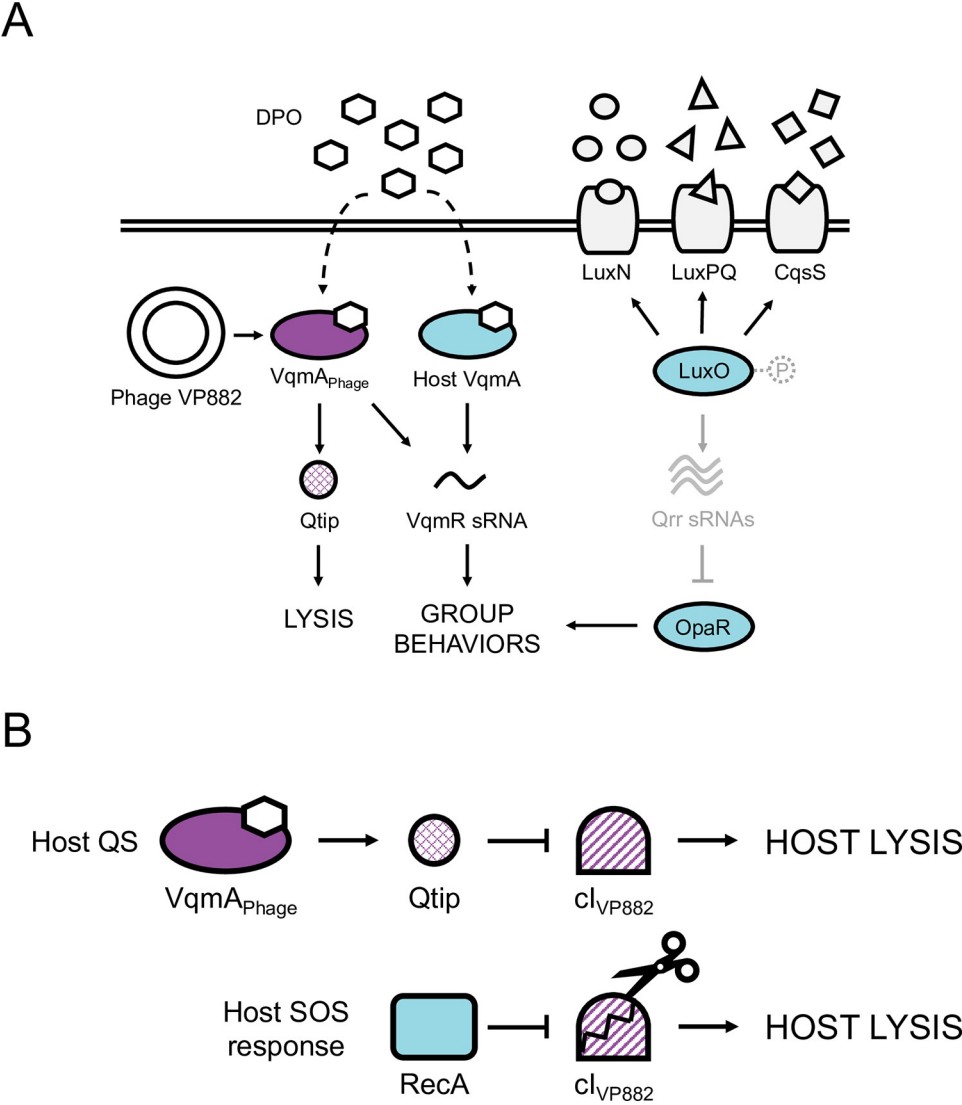

**Fig 1. Simplified schematics of the vibriophage VP882 and *Vibrio* QS circuits at HCD and mechanisms driving phage VP882 host-cell lysis.** (A) Multiple autoinducer-receptor pairs control QS in *V. parahaemolyticus*. AI-1 (circles), AI-2 (triangles), and CAI-1 (squares) are detected by LuxN, LuxPQ, and CqsS, respectively. At LCD, the receptors are kinases and funnel phosphate to LuxO (via a protein LuxU, not shown). LuxO~P activates transcription of genes encoding the Qrr sRNAs. The Qrr sRNAs post-transcriptionally repress *opaR*, the master regulator of group behaviors. At HCD (shown in the cartoon), the liganded receptors act as phosphatases, reversing the phosphorylation cascade. Consequently, production of Qrr sRNAs ceases, and *opaR* translation is derepressed, OpaR is produced, and the cells undertake group behaviors. In parallel, at HCD, the DPO autoinducer (hexagons) interacts with the VqmA transcription factor. The complex activates *vqmR* expression and the VqmR sRNA controls QS behaviors. Phage VP882 carries *vqmA*~Phage~. Binding of VqmA~Phage~ to host-produced DPO activates expression of a gene encoding an antirepressor called Qtip, which drives the phage lytic pathway. VqmA~Phage~ also binds the host *vqmR* promoter to drive host QS behaviors. (B) Two pathways control phage VP882 lysis-lysogeny transitions. Upper: DPO-bound VqmA~Phage~ activates expression of *qtip*. Qtip sequesters the phage cI~VP882~ repressor of lysis leading to phage replication and host-cell killing. Lower: the phage cI~VP882~ repressor is proteolyzed in response to host SOS/DNA damage via a host RecA-dependent mechanism. The consequence is phage replication and host-cell killing.

proteolysis of cI~VP882~ (Fig 1B). Sequestration or proteolysis of cI~VP882~ derepresses the phage *q* gene. Q activates expression of the *gp69-71* operon encoding the phage lytic genes, driving host-cell lysis. Presumably, tuning into the host SOS response allows phage VP882 to connect

its lifestyle transitions to host viability, whereas tuning into host QS allows phage VP882 to execute host-cell lysis at high host-cell density when the probability of infecting the next host cell is maximized [8].

*V. parahaemolyticus*, like all studied *Vibrios*, harbors multiple QS circuits that function in parallel (Fig 1A and [12]). In addition to DPO, *V. parahaemolyticus* produces and detects three other AIs: AI-1 (*N*-(3-hydroxybutanoyl)-L-homoserine lactone), AI-2 ((2*S*,4*S*)-2-methyl-2,3,3,4-tetrahydroxytetrahydrofuran-borate), and CAI-1 ((*Z*)-3-aminoundec-2-en-4-one), and they are detected by the membrane-bound QS receptors LuxN, LuxPQ, and CqsS, respectively (Fig 1A and [12]). At low cell density (LCD), unliganded LuxN, LuxPQ, and CqsS act as kinases and shuttle phosphate to the LuxO QS transcriptional regulator. Phosphorylated LuxO (LuxO~P) activates expression of genes encoding five sRNAs, called Qrr1-5, that repress translation of *opaR*, encoding the QS high cell density (HCD) master regulator OpaR. At HCD, LuxN, LuxPQ, and CqsS bind their partner AIs, the receptors convert to phosphatases, and the flow of phosphate through the cascade is reversed. LuxO is dephosphorylated and inactivated. Under this condition, *qrr* genes are not expressed, allowing OpaR production, and consequently, collective behaviors (Fig 1A).

Here, we report the genome sequence of *V. parahaemolyticus* strain O3:K6 882 (here forward called strain 882), the strain from which phage VP882 was isolated. Strikingly, strain 882 encodes *vqmA* (*vqmA*$_{882}$) but lacks its partner *vqmR* gene. Specifically, there is an 897 bp deletion relative to other *V. parahaemolyticus* strains that eliminates the *vqmR* promoter, the *vqmR* coding sequence, and a portion of the non-coding region upstream of *vqmA*$_{882}$. We also discover that the strain 882 *luxO* gene (*luxO*$_{882}$) harbors a 36 bp deletion that leads it to function as a phospho-mimetic. In this report, we investigate the consequences of these mutations on the QS capabilities of strain 882 and the broader implications for host-phage biology. We show that lack of *vqmR* and *vqmA*$_{882}$ expression, and the phospho-mimetic allele of *luxO*$_{882}$ lock strain 882 into the LCD QS state. Repair of *vqmR*-*vqmA*$_{882}$ or *luxO*$_{882}$ or both *vqmR*-*vqmA*$_{882}$ and *luxO*$_{882}$ partially and fully, respectively, restores QS capability. Re-establishment of strain 882 QS capability activates phage VP882 gene expression, more rapid phage VP882-driven host-cell lysis, and production of higher titers of virions compared to that in the QS-incompetent parent strain. We propose that the locked LCD QS state silences the information encoded in QS AIs for both the strain 882 host and the VP882 phage. This arrangement suppresses the potential for phage VP882 to transition from lysogeny to lysis, presumably protecting the host from killing by the phage that it carries.

## Results

### V. parahaemolyticus *strain 882 lacks* vqmR *and does not express* vqmA

Strain 882 is unique relative to other studied *V. parahaemolyticus* strains because it harbors a phage with the capacity to surveil host QS by monitoring DPO abundance. The DNA sequence of phage VP882 is deposited [6], however, no sequence of its host was reported. We sequenced the strain 882 genome and compared it to that of the closely related *V. parahaemolyticus* O3:K6 type-strain, RIMD2210633. Strain 882 has an 897 bp deletion that eliminates its *vqmR* gene and promoter, a portion of a neighboring upstream gene (a putative transcriptional regulator of a proline metabolic operon), a predicted gene of unknown function, and a portion of the non-coding region upstream of *vqmA*$_{882}$ (Fig 2A). To determine the incidence of the *vqmR* deletion, we examined the coding regions of *vqmR*-*vqmA* loci in other publicly available *Vibrio* DNA sequences (5460 genomes total). We disregarded 12 of the 5460 *Vibrio* sequences whose assembly gaps included this region. Most sequenced *Vibrios* encode predicted *vqmR*-*vqmA* pairs. However, 312 strains lack both *vqmR* and *vqmA* (S1A Fig), and an additional 37 strains

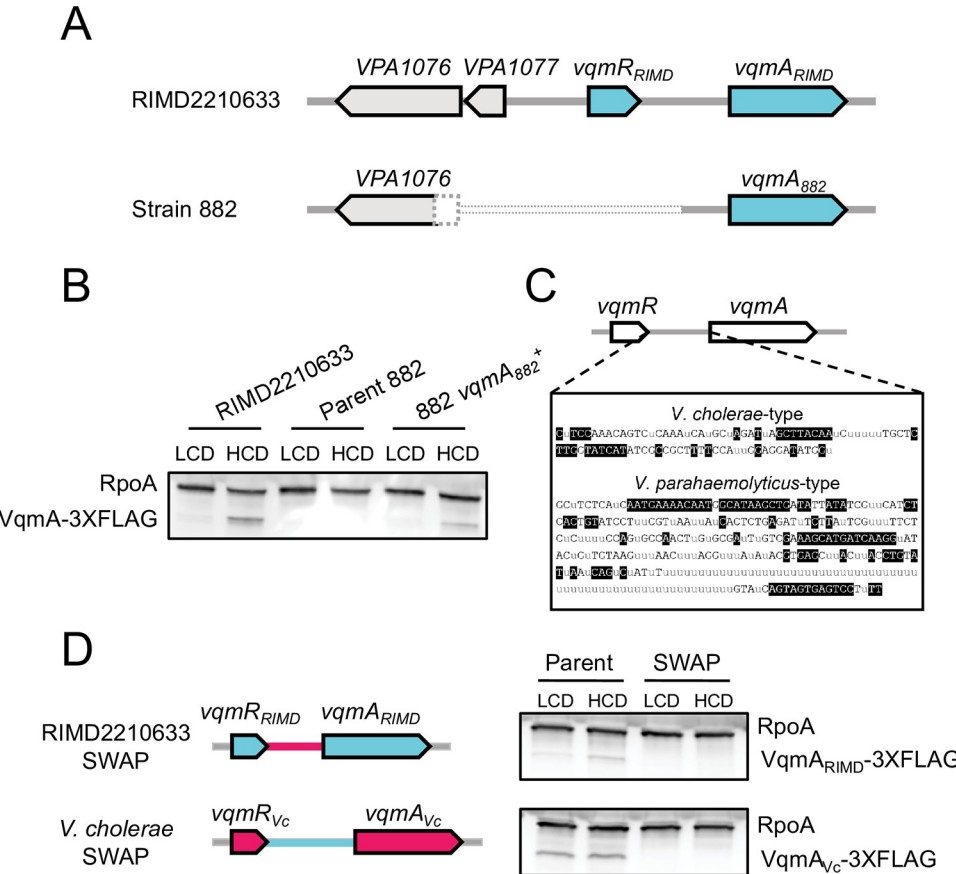

**Fig 2. VqmA and VqmR are not produced in strain 882.** (A) Schematic of the *vqmR-vqmA* loci of
*V. parahaemolyticus* RIMD2210633 and strain 882. The dotted pattern signifies that strain 882 harbors an 897 bp
deletion encompassing a portion of a neighboring upstream gene (*VPA1076*, encoding a putative transcriptional
regulator of a proline metabolic operon), a putative gene of unknown function (*VPA1077*), the *vqmR* promoter, the
*vqmR* coding sequence, and a portion of the non-coding region upstream of *vqmA_{882}*. (B) Representative western blot
showing VqmA_{RIMD}-3XFLAG in *V. parahaemolyticus* RIMD2210633, VqmA_{882}-3XFLAG in strain 882, and
VqmA_{882}-3XFLAG in strain 882 with the non-coding region upstream of *vqmA_{882}* restored (i.e., this strain is 882
*vqmA_{882}*^+). RpoA was used as the loading control. (C) Consensus sequences of the intergenic regions between *vqmR*
and *vqmA* for the two designated groups. Gray "u" letters indicate ≤50% agreement. Black nucleotides with clear
backgrounds indicate >50% but <100% agreement. White nucleotides with black backgrounds indicate 100%
agreement. (D) *Left*: schematic showing exchange of regions between *V. parahaemolyticus* RIMD2210633 and
*V. cholerae* at the *vqmR-vqmA* loci. Genes and non-coding regions from *V. parahaemolyticus* RIMD2210633 and
*V. cholerae* are colored cyan and red, respectively. *Right*: representative western blot of VqmA-3XFLAG produced by
*V. parahaemolyticus* RIMD2210633 and *V. cholerae* from their native promoters (Parent) and following exchange of
their promoters (SWAP). RpoA was used as the loading control. Data are representative of three independent
experiments (B) and two independent experiments (D).

possess either a frameshift or a nonsense mutation in *vqmA*, presumably disrupting its func-
tion. Regarding *V. parahaemolyticus* strains (1731), all possess *vqmR-vqmA*. In three cases,
however, the *vqmA* gene is predicted to be non-functional (frameshift or nonsense mutation).
From this analysis, we could not discern if nucleotide changes in *vqmR* genes affect the func-
tions or the expression levels of the VqmR sRNAs. Interestingly, of the >5000 analyzed strains,
strain 882 is the only sequenced *Vibrio* that lacks *vqmR* but retains *vqmA* in its genome.

Our findings suggest that in strain 882, the VqmA-VqmR QS circuit cannot function
because *vqmR* is absent. Moreover, given that the deleted region includes the predicted *vqmA_{882}*
promoter, it was not clear if *vqmA_{882}* is expressed. Western blot analysis of VqmA_{882}-3XFLAG

produced from the native locus shows that VqmA$_{882}$ is not made (Fig 2B). By contrast, *V. parahaemolyticus* RIMD2210633 harbors an intact *vqmR-vqmA* locus and VqmA$_{RIMD}$-3XFLAG was produced, and at higher levels at HCD than at LCD (Fig 2B). Thus, the non-coding region deleted in strain 882 contains the *vqmA$_{882}$* promoter, which eliminates *vqmA$_{882}$* expression. To test this prediction, we repaired the putative *vqmA$_{882}$* promoter region by inserting the analogous intergenic DNA from *V. parahaemolyticus* RIMD2210633. VqmA$_{882}$ production was restored despite the absence of *vqmR* (Fig 2B). We call this strain 882 *vqmA$_{882}$*$^{+}$.

Curiously, we observed that *vqmA* expression is cell-density dependent in both *V. parahaemolyticus* RIMD2210633 and strain 882 *vqmA$_{882}$*$^{+}$. In *V. cholerae* strain C6706, the only species in which *vqmA* expression has been studied, *vqmA$_{Vc}$* is expressed and VqmA$_{Vc}$ is made constitutively [7]. We eliminated the possibility that *V. parahaemolyticus* VqmA directly activates its own expression (S1B Fig). To assess the underlying mechanism for the species-specific difference in expression patterns, we aligned *Vibrio vqmA* promoters (S1C Fig), revealing two potential clades differing by length and nucleotide sequence (Fig 2C). The first group is comprised of shorter, *V. cholerae*-type *vqmA* promoters (~100 bp, Figs 2C and S1C), and the second is comprised of longer, *V. parahaemolyticus* RIMD2210633-type *vqmA* promoters (~300 bp; Figs 2C and S1C). Exchanging *V. cholerae* and *V. parahaemolyticus* RIMD2210633 *vqmA* promoters eliminated production of both VqmA proteins as judged by Western blot analysis (Fig 2D). Thus, the *vqmA* promoter sequence predicts whether *vqmA* expression is constitutive, as in *V. cholerae*, or cell-density dependent, as in *V. parahaemolyticus* RIMD2210633, and the promoter sequence only functions in its species of origin.

We wondered whether VqmA$_{882}$, if produced by strain 882, would bind DPO and activate expression of *vqmR*. The *vqmA$_{882}$* coding sequence is identical to those of other *V. parahaemolyticus* strains, including *V. parahaemolyticus* RIMD2210633. VqmA$_{882}$ also shares 65% and 41% amino acid identity with VqmA$_{Vc}$ and VqmA$_{Phage}$, respectively, and, residues F67, F99, and K101, which are critical for DPO binding in VqmA$_{Vc}$ and VqmA$_{Phage}$, are conserved in VqmA$_{882}$ (S2A Fig and [13]). We also wondered whether VqmA$_{882}$ operates analogously to VqmA$_{Vc}$ in that it does not bind the phage *qtip* promoter. To examine VqmA promoter binding capabilities, which have not previously been studied for *V. parahaemolyticus*, we transformed arabinose-inducible *vqmA$_{882}$* into *Escherichia coli* harboring P*vqmR$_{RIMD}$* or P*qtip* fused to the luciferase genes (P*vqmR$_{RIMD}$-lux* or P*qtip-lux*, respectively). S2B Fig shows that, in the presence of DPO, induction of VqmA$_{882}$ drove a ~500-fold increase in P*vqmR$_{RIMD}$-lux* activity, and maximum light production depended on DPO being supplied (S2C Fig). By contrast, a <5-fold change in P*qtip-lux* occurred (S2B Fig). Thus, the *vqmA$_{882}$* gene, if expressed in strain 882, would function analogously to VqmA$_{Vc}$ in *V. cholerae*. However, the VqmR-VqmA QS system in strain 882 is defective for two reasons–because it lacks *vqmR* and it does not express *vqmA$_{882}$*.

## *V. parahaemolyticus* strain 882 possesses a LCD-locked LuxO allele

Inspired by our finding that in strain 882, the VqmA-VqmR QS circuit is non-functional, we determined whether its other QS systems also harbored defects. Examination of QS genes revealed a 36 bp deletion exists in *luxO* (*luxO$_{882}$*) corresponding to elimination of LuxO residues 91–102 (Δ91–102, and Fig 3A). To characterize the consequence of the deletion on LuxO function, we introduced the QS-controlled luciferase operon from *Vibrio harveyi* (*luxCDABE*) into strain 882 and used light production as a heterologous QS target. In *V. harveyi* at HCD, *luxCDABE* expression is activated by LuxR, the HCD QS master regulator [12]. Across *Vibrios*, homologs of LuxR including OpaR from *V. parahaemolyticus*, also activate *luxCDABE* at HCD [14]. We compared light production by strain 882 to three other 882 strains that we constructed:

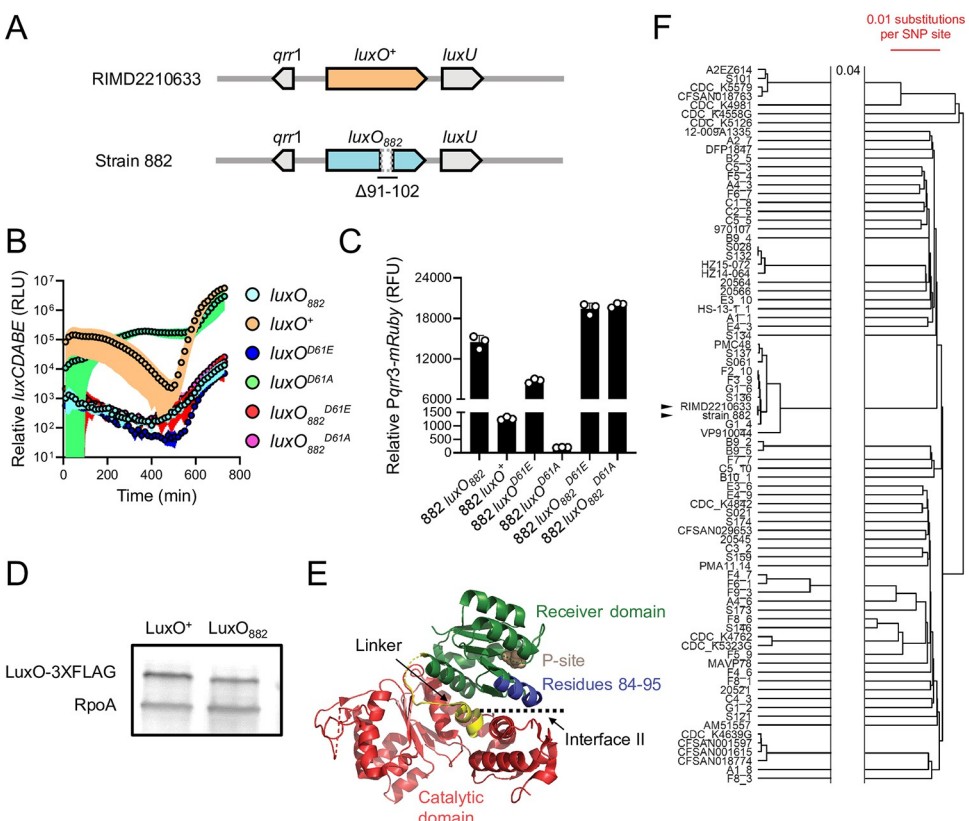

**Fig 3. LuxO₈₈₂ harbors a 12 amino acid deletion, rendering the protein LCD-locked.** (A) Schematic of the *luxO*-containing loci of *V. parahaemolyticus* RIMD2210633 and *V. parahaemolyticus* strain 882. Strain 882 *luxO* harbors a 36 bp deletion corresponding to residues 91–102, as indicated. (B) Relative *luxCDABE* output over time from the parent strain 882 (*luxO₈₈₂*; cyan), strain 882 *luxO*⁺ (orange), strain 882 *luxO^{D61E}* (blue), strain 882 *luxO^{D61A}* (green), strain 882 *luxO₈₈₂^{D61E}* (red), and strain 882 *luxO₈₈₂^{D61A}* (purple). RLU designates relative light units. As noted in the main text, all *luxO₈₈₂* alleles harbor the 12 amino acid Δ91–102 deletion that is not present in the RIMD2210633 *luxO* alleles. (C) Relative P*qrr3-mRuby* output for the indicated 882 strains. RFU designates relative fluorescence units. (D) Representative western blot of LuxO⁺-3XFLAG and LuxO₈₈₂-3XFLAG produced by strain 882. RpoA was used as the loading control. (E) Crystal structure of *V. angustum* LuxO (PDB: 5EP0 and [16]). The receiver, catalytic, and linker domains are colored in green, red, and yellow, respectively. The aspartate site of phosphorylation is shown in brown (denoted P-site). Residues 84–95 in *V. angustum* LuxO, corresponding to residues 91–102 in *V. parahaemolyticus* LuxO, which are missing in *V. parahaemolyticus* strain 882 LuxO, are colored in blue. Interface II is indicated. (F) A phylogenetic tree constructed using UPGMA based on core genome SNP alignment of the 88 *V. parahaemolyticus* strains carrying the Δ91–102 LuxO mutation. Scale bar (red) indicates 0.01 nucleotide substitution per SNP site. The two horizontal lines indicate a gap in the tree, and 0.04 refers to nucleotide substitutions per SNP site separating the gap. Black arrowheads show strain 882 and *V. parahaemolyticus* RIMD2210633. Data are represented as means ± std with *n* = 3 biological replicates (B, C) and representative of three independent experiments (D).

*luxO*⁺ (in which *luxO₈₈₂* was repaired), *luxO^{D61E}*, and *luxO^{D61A}*. LuxO D61E is a LuxO~P mimetic and drives constitutive production of the Qrr sRNAs, thus locking cells into the LCD QS state [15]. Conversely, LuxO D61A is a non-phosphorylatable LuxO allele, is inactive, and therefore locks cells into the HCD QS state. As expected, the 882 strains carrying *luxO^{D61E}* and *luxO^{D61A}* were constitutively dark and bright, respectively (Fig 3B and [15]), and strain 882 engineered to be *luxO*⁺ produced light in a cell density-dependent manner, the hallmark of a QS-controlled process. By contrast, the parent strain 882 produced a modest amount of light (~100-1000-fold less than the 882 *luxO*⁺ strain at HCD) with dramatically reduced cell density-dependent regulation (Fig 3B). The light production pattern of strain 882 was most similar to that of strain 882 carrying *luxO^{D61E}* (Fig 3B), indicating that LuxO₈₈₂ mimics LuxO~P

(see Fig 1A). Indeed, introduction of a reporter of *qrr3* expression (P*qrr3-mRuby*) verified that both *luxO₈₈₂* and *luxO^{D61E}* drove high levels of fluorescence, whereas *luxO⁺* and *luxO^{D61A}* produced 11-fold and 73-fold less fluorescence, respectively (Fig 3C). Moreover, because repair of *luxO₈₈₂* to *luxO⁺* fully restored proper QS function, the deletion in *luxO₈₈₂* is the defect that eliminates signal transduction from LuxN, LuxPQ, and CqsS (Fig 1A and [12]). Taken together, our results show that strain 882 is LCD-locked with respect to all of its known QS systems.

The LCD-behavior driven by *luxO₈₈₂* could be due to increased LuxO transcriptional activity, increased LuxO protein production, or both. First, to examine LuxO activity, we engineered the D61E and D61A mutations into *luxO₈₈₂* harboring the 36 bp deletion and introduced the alleles onto the strain 882 chromosome. We call these genes *luxO₈₈₂^{D61E}* and *luxO₈₈₂^{D61A}*, corresponding to the Δ91–102 LuxO D61E and Δ91–102 LuxO D61A proteins, respectively. We assayed activity using the *luxCDABE* reporter. As expected, strain 882 with *luxO₈₈₂^{D61E}* was constitutively dark (Fig 3B). However, in contrast to the constitutively bright phenotype of strain 882 harboring *luxO^{D61A}*, strain 882 carrying *luxO₈₈₂^{D61A}* was also constitutively dark (Fig 3B). Consistent with this result, both *luxO₈₈₂^{D61E}* and *luxO₈₈₂^{D61A}* drove high *qrr*3 expression (Fig 3C). Thus, the Δ91–102 mutation in LuxO₈₈₂ eliminates the requirement for phosphorylation for LuxO to possess transcriptional activity. Second, to examine whether the Δ91–102 deletion changes LuxO protein production, we engineered *luxO⁺-3XFLAG* and *luxO₈₈₂-3XFLAG* alleles into the chromosome of strain 882 and verified that the tagged proteins function like untagged LuxO⁺ and LuxO₈₈₂, respectively (S3A Fig). Western blot analyses revealed that LuxO⁺ and LuxO₈₈₂ are produced at similar levels, showing that increased LuxO₈₈₂ protein levels do not underlie its LCD-locked phenotype (Fig 3D). Finally, mapping sequences onto the existing LuxO crystal structure revealed that the residues missing in LuxO₈₈₂ reside along interface II between the LuxO receiver domain which contains the D61 site of phosphorylation and the ATP-binding catalytic domain (Fig 3E). Phosphorylation of LuxO is predicted to disrupt interface II, thereby enhancing LuxO transcriptional activity [16]. We predict that deletion of residues 91–102 alters interface II, and consequently, increases LuxO activity.

To investigate whether other *Vibrios* possess *luxO* mutations similar to that present in strain 882, we analyzed the *luxO* genes of the same 5460 sequenced *Vibrios* described above and found that 88 *Vibrio* strains, including strain 882, carry the identical Δ91–102 *luxO* mutation (S3B Fig). This group of *Vibrios* is dominated by *V. parahaemolyticus* strains. Five *V. cholerae* and three *Vibrio owensii* strains are also in the group. Other large, in-frame deletions are present in *luxO* genes in several *V. parahaemolyticus* strains spanning residues 67–87 and 114–134 (S3C Fig). While we did not test the functions of these LuxO proteins, we speculate that they act similarly to LuxO₈₈₂. Regarding the frequency of the Δ91–102 *luxO* mutation in *V. parahaemolyticus* strains, several possibilities could explain our finding: (1) There is an overrepresentation of sequenced *V. parahaemolyticus* strains in the database (1731/5460), (2) the mutation originated from a common ancestor, and/or (3) environmental pressures on *V. parahaemolyticus* strains select for this particular mutation. We cannot eliminate the first possibility of sample bias. To distinguish between the latter two possibilities, we performed a core genome alignment of the 88 *V. parahaemolyticus* strains carrying Δ91–102 *luxO* (including strain 882) with *V. parahaemolyticus* RIMD2210633 as the wild-type reference (Fig 3F). Our analyses suggest that the incidence of the Δ91–102 *luxO* mutation does not reflect acquisition from a common ancestor, given that strain 882 is most similar to *V. parahaemolyticus* RIMD2210633 yet the two have dissimilar *luxO* alleles. Rather, it is more likely that *V. parahaemolyticus* strains encounter environmental pressures that select for mutations in QS components that maintain the strains in the LCD QS mode.

## V. parahaemolyticus *strain 882 QS circuits converge to regulate phage VP882 gene expression by an SOS-independent mechanism*

One of our goals is to understand how QS shapes bacterial-phage partnerships. In the present context, we sought to determine the ramifications the QS defects have on interactions between strain 882 and its lysogenized prophage VP882. Toward this aim, we assessed the transcriptional changes that occur in strain 882 following repair of the *vqmR-vqmA* and *luxO* QS pathways, individually and together. We call the new strains 882 *vqmR-vqmA$_{882}$$^+$*, 882 *luxO$^+$*, and 882 *vqmR-vqmA$_{882}$$^+$ luxO$^+$*. Strikingly, genes encoded by phage VP882 were upregulated 2-10-fold in 882 *luxO$^+$* and 3-15-fold in 882 *vqmR-vqmA$_{882}$$^+$ luxO$^+$* relative to strain 882 (Fig 4A and S1 Table). Specifically, transcription of 40/71 predicted genes on phage VP882 increased in the repaired strains indicating that proper host QS function promotes phage VP882 gene expression. Moreover, the 882 *vqmR-vqmA$_{882}$$^+$ luxO$^+$* strain displayed the highest activation of phage VP882 genes, suggesting that VqmR-VqmA$_{882}$ and LuxO function additively to control phage genes. Consistent with these results, compared to strain 882, there was no significant increase in expression of phage VP882 genes in the 882 *vqmR-vqmA$_{882}$$^+$* strain carrying *luxO$_{882}$* (Fig 4A).

A transcriptional reporter in which the phage *gp69* lytic gene promoter was fused to *lux* (P*gp69-lux*) verified that phage VP882 gene expression is activated by host QS. Specifically: the 882 *luxO$^+$* and 882 *vqmR-vqmA$_{882}$$^+$ luxO$^+$* strains exhibited ~10-20-fold higher P*gp69-lux* expression than strains 882 and 882 *vqmR-vqmA$_{882}$$^+$* (Fig 4B). Light production from P*gp69-lux* was nearly undetectable in 882 strains that had been cured of phage VP882, indicating that the QS state of the host affects P*gp69* activity via phage-encoded regulators (Fig 4C). Indeed, the transcriptomic analyses show that the key phage lytic genes *vqmA$_{Phage}$*, *qtip*, and *q* were upregulated in 882 *luxO$^+$* and 882 *vqmR-vqmA$_{882}$$^+$ luxO$^+$*, while expression of the *cI$_{VP882}$* gene remained unchanged (Fig 4A and S1 Table).

There were no significant growth differences among 882 test strains cultivated under standard liquid growth conditions (S4A Fig). This finding suggests that, while host QS activates phage VP882 lytic gene expression, the level of activation is insufficient to produce global host-cell lysis. Thus, to characterize the consequences of host QS signaling on phage VP882 lifestyle transitions, we monitored two key features of phage virulence, phage reproductive success (i.e., virion production) and time to host-cell lysis following phage induction in each of our test strains. First, regarding QS effects on spontaneous virion production, consistent with the above P*gp69-lux* reporter data, the QS repaired 882 *luxO$^+$* and 882 *vqmR-vqmA$_{882}$$^+$ luxO$^+$* strains produced twice as many viral particles as the QS-defective parent strain 882 despite there being no detectable differences in growth under our conditions (Figs 5A, *left*, and S4A). Second, regarding QS effects on time to lysis, we introduced a plasmid carrying arabinose-inducible *vqmA$_{Phage}$* into each strain to induce the phage lytic program and drive host lysis. Strains 882 *luxO$^+$* and 882 *vqmR-vqmA$_{882}$$^+$ luxO$^+$* lysed 1 and 2 h earlier, respectively, than did strains 882 and 882 *vqmR-vqmA$_{882}$$^+$* (Fig 5B). Similar to the results for spontaneous induction of the phage, following expression of *vqmA$_{Phage}$*, the 882 *luxO$^+$* and 882 *vqmR-vqmA$_{882}$$^+$ luxO$^+$* strains produced twice as many viral particles as strain 882 (Fig 5A, *right*).

We considered two possible mechanisms by which host QS could affect phage VP882 lytic activity: Host QS could increase RecA-dependent cI$_{VP882}$ proteolysis or host QS could activate phage-encoded *qtip* expression. To test the first possibility, we introduced HALO-tagged cI$_{VP882}$ (*HALO-cI$_{VP882}$*) into our set of 882 lysogens and monitored cI$_{VP882}$ cleavage. No differences were detected between the strains in the absence of ciprofloxacin, which measures spontaneous HALO-cI$_{VP882}$ proteolysis. In the presence of ciprofloxacin, which induces HALO-cI$_{VP882}$ cleavage, all strains exhibited similar elevations in proteolyzed HALO-cI$_{VP882}$

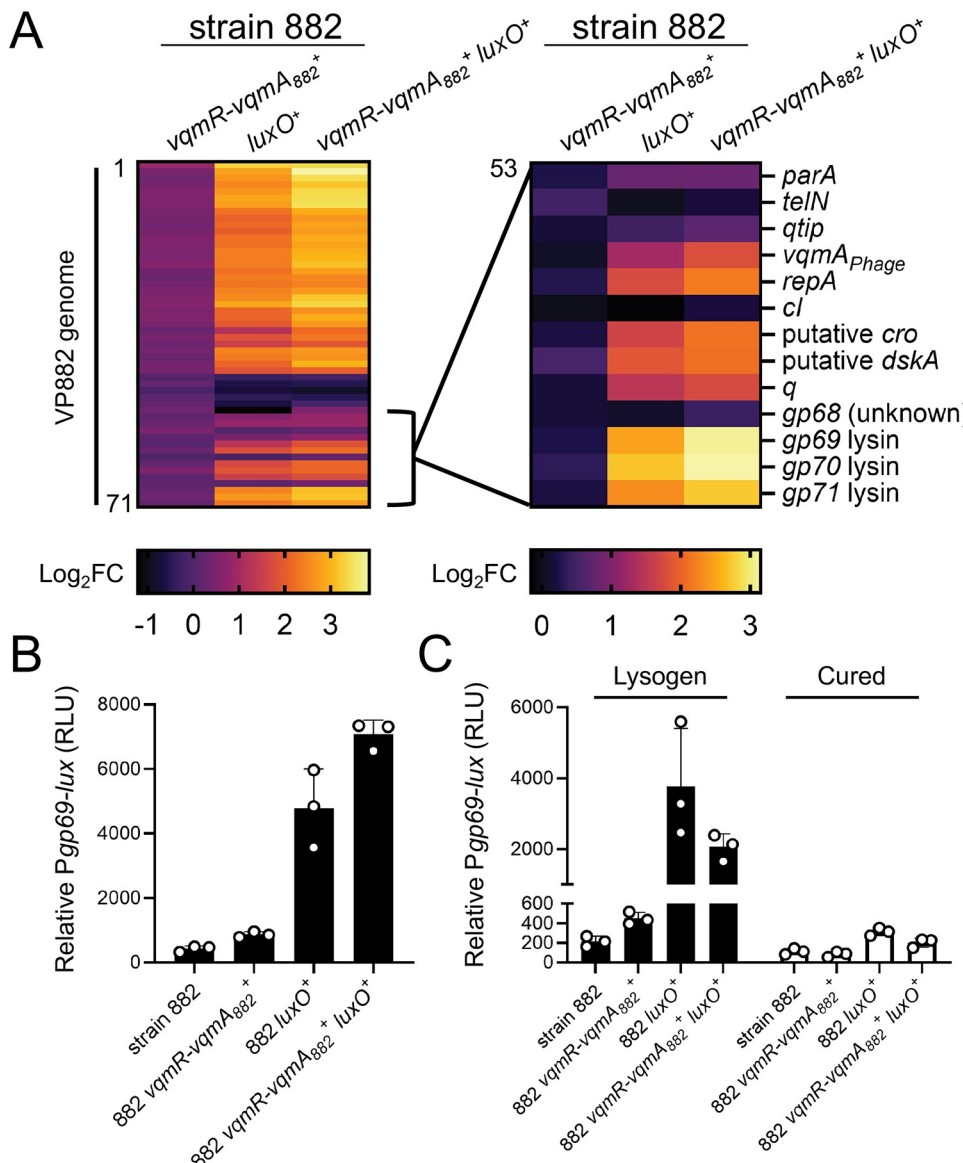

**Fig 4. Restoration of host QS function activates phage VP882 gene expression.** (A) *Left*: heatmap showing log$_2$ fold changes (FC) of all quantified genes in phage VP882 (*gp1-71*) in the indicated 882 strains. Data are normalized to the parent 882 strain. Numerical values for heatmaps are available in S1 Table. Increasing brightness represents increasing gene expression. *Right*: as in the left panel, heatmap showing log$_2$ fold changes of the indicated phage VP882 genes (*gp53-71*). (B) Relative P*gp69-lux* output in the indicated 882 strains. (C) Relative P*gp69-lux* output in the indicated 882 strains carrying either phage VP882 (lysogen; black bars) or lacking phage VP882 (cured; white bars). Data are represented as means with *n* = 3 biological replicates (A), and as means ± std with *n* = 3 biological replicates (B, C). RLU as in Fig 2B (B, C).

(Fig 6A). Consistent with this finding, deletion of *recA* in the 882 lysogens did not eliminate differences in P*gp69-lux* expression or time-to-lysis between the QS-repaired and QS-defective 882 strains (Fig 6B and 6C, respectively). Thus, host QS input into phage VP882 lifestyle transitions occurs independently of the RecA pathway.

To test the possibility that host QS increases Qtip-mediated cI$_{VP882}$ sequestration, we constructed phage VP882 *qtip::cm*, eliminating the phage VqmA$_{Phage}$-Qtip pathway to lysis (S4B Fig). We assayed P*gp69-lux* expression in our set of QS-repaired and QS-defective 882

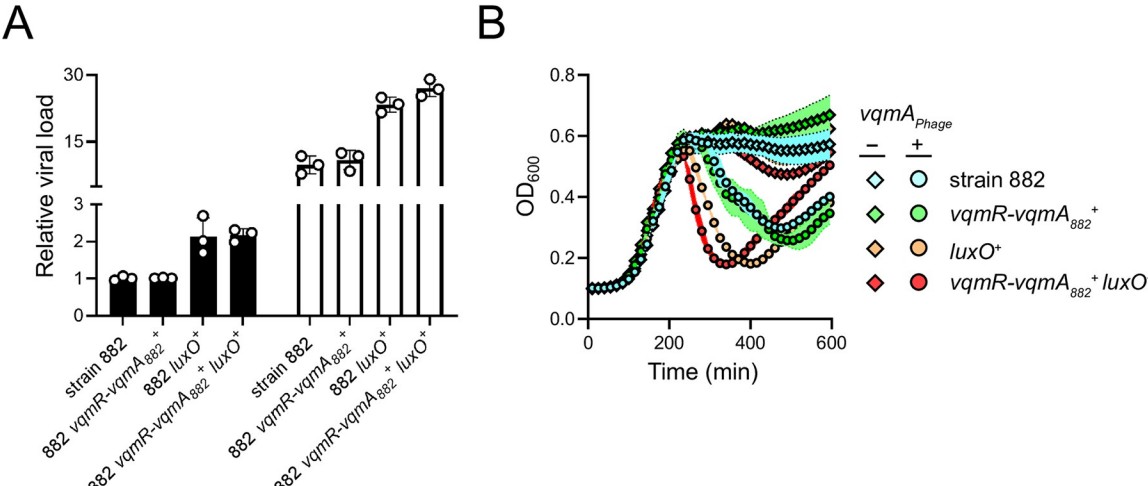

**Fig 5. Phage VP882 is more virulent in QS-competent *V. parahaemolyticus* than in QS-deficient *V. parahaemolyticus*.** (A) Quantitation of viral particles collected from the indicated 882 strains carrying arabinose-inducible *vqmA$_{Phage}$* grown in medium lacking (black bars) or containing 0.02% arabinose (white bars). Relative viral load is the amount of phage VP882-specific DNA (*gp69*) relative to non-phage DNA (*hfq*) measured by qPCR. (B) Growth of the 882 parent (cyan), 882 *vqmR-vqmA$_{882}$$^+$* (green), 882 *luxO$^+$* (orange), and 882 *vqmR-vqmA$_{882}$$^+$ luxO$^+$* (red) strains carrying arabinose-inducible *vqmA$_{Phage}$* and grown in medium lacking (diamonds) or containing (circles) 0.02% arabinose. Data are represented as means ± std with $n = 3$ biological replicates and $n = 4$ technical replicates (A), and as means ± std with $n = 3$ biological replicates (B).

strains carrying phage VP882 *qtip*::*cm* or a phage we call VP882 Ctr::*cm* with *cm* integrated at a neutral locus. Interruption of *qtip* severely reduced host QS-dependent effects on P*gp69-lux* expression (Fig 6D), indicating that host QS influences phage lysis-lysogeny transitions via a Qtip-dependent mechanism. As the only known regulator of *qtip* is VqmA$_{Phage}$, we tested whether a fragment of the phage VP882 genome harboring *vqmA$_{Phage}$-qtip* and the key *cI$_{VP882}$* and *q* regulatory genes is sufficient to drive differential P*gp69-lux* induction in the various test strains. Similar to the 882 strains carrying full-length phage VP882, the 882 *luxO$^+$* and 882 *vqmR-vqmA$^+$ luxO$^+$* strains harboring only the phage VP882 regulatory fragment (called VP882*) produced >10-fold more light from P*gp69-lux* than did strains 882 and 882 *vqmR-vqmA$^+$* carrying VP882* (Fig 6E, *left*). Deletion of *vqmA$_{Phage}$-qtip* from VP882* (VP882* Δ*vqmA$_{Phage}$-qtip*) dramatically reduced P*gp69-lux* levels in all the strains, indeed, abolishing differences between them (Fig 6E, *right*). We conclude that regulation of the phage-encoded *vqmA$_{Phage}$-qtip* pathway connects host QS to phage VP882 lytic gene expression.

## Discussion

In this report, we characterize *V. parahaemolyticus* strain 882, the natural host of phage VP882. We discover that strain 882 harbors defects in all of its QS circuits, such that the strain is locked in the LCD QS mode. The consequence is suppression of phage VP882 lytic activation. Thus, phage VP882 is more virulent to QS-proficient *V. parahaemolyticus* 882 than to the naturally occurring QS-deficient 882 strain. We previously showed that, at HCD, detection of host-produced DPO by phage VP882 launches phage-mediated bacterial killing [8]. Our present results suggest that phage VP882 reacts to inputs from multiple QS components: DPO, VqmR-VqmA, and LuxO.

We do not yet know the mechanism by which LuxO and VqmR-VqmA funnel information into activation of phage VP882 lytic genes. We do know that VqmA$_{Phage}$ and Qtip are required, and that the RecA-dependent pathway is dispensable. We presume that the host QS

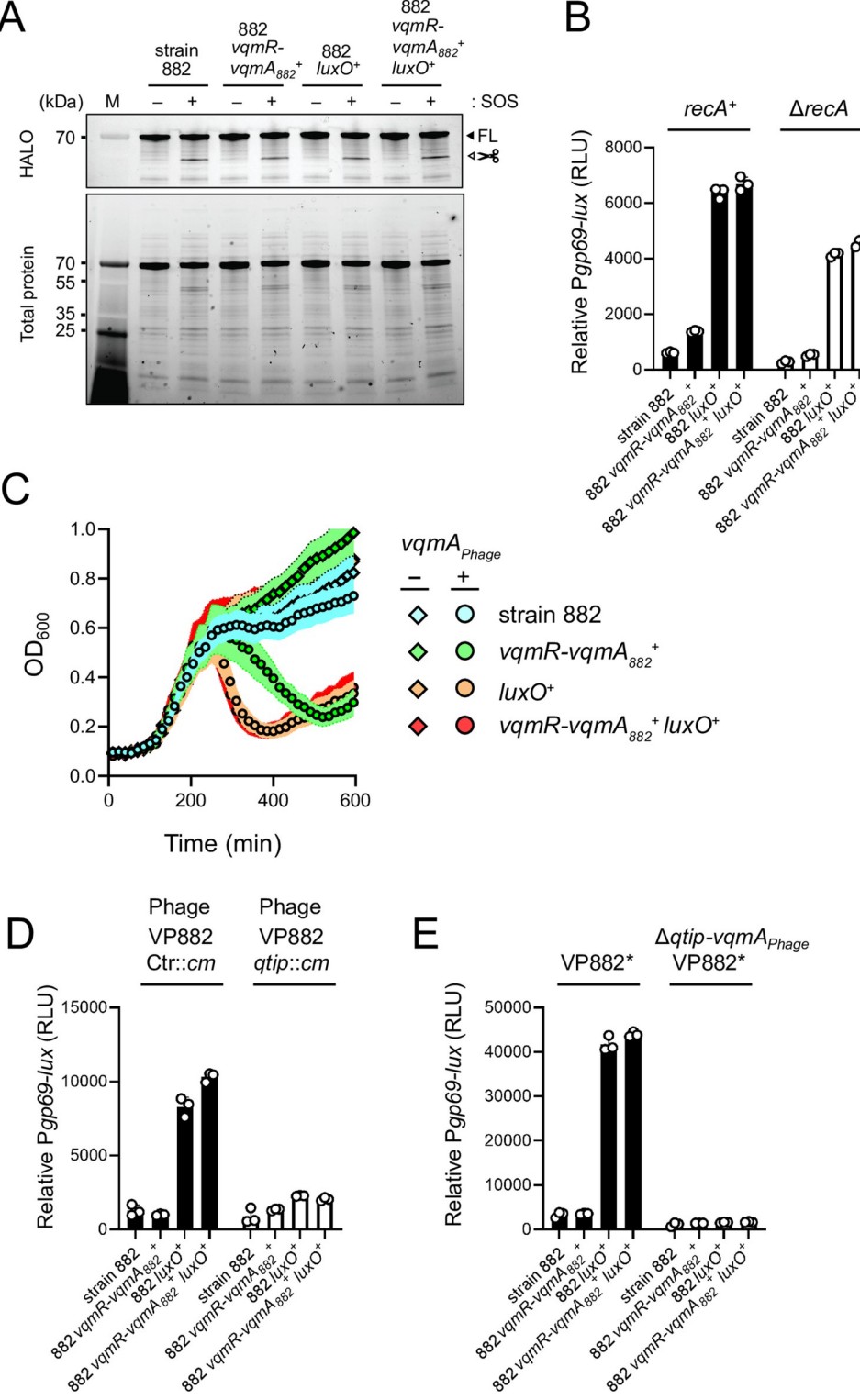

**Fig 6. Qtip-mediated inactivation of cI$_{VP882}$, and not RecA-mediated cI$_{VP882}$ proteolysis, is required for host QS control of phage VP882 gene expression.** (A) SDS-PAGE in-gel labeling of HALO-cI$_{VP882}$ produced from a plasmid in the indicated strains. - and + indicate, respectively, the absence and presence of ciprofloxacin to induce SOS. The FL and scissors labels designate full-length and cleaved cI$_{VP882}$ proteins, respectively. M denotes the molecular weight marker. kDa is kilodaltons. (B) Relative P*gp69-lux* output in the indicated *recA*$^+$ (black bars) and Δ*recA* (white bars) 882 strains. (C) Growth of Δ*recA* strain 882 (cyan), Δ*recA* 882 *vqmR-vqmA*$_{882}$$^+$ (green), Δ*recA* 882 *luxO*$^+$ (orange), and

$\Delta recA$ 882 *vqmR-vqmA$_{882}^{+}$ luxO$^{+}$* (red) strains carrying arabinose-inducible *vqmA$_{Phage}$* and grown in medium lacking (diamonds) or containing (circles*)* 0.02% arabinose. (D) As in (B), except with strains carrying either phage VP882 Ctr::*cm* (black bars) or phage VP882 *qtip*::*cm* (white bars). All strains are *recA$^{+}$*. (E) As in (B), except with strains carrying either VP882* or VP882* $\Delta vqmA_{Phage}$-*qtip*. All strains are *recA$^{+}$*. Data are represented as means ± std with $n$ = 3 biological replicates (B, C, D, E), and as representative of three independent experiments (A). RLU as in Fig 2B (B, D, E).

circuits converge on regulation of transcription of *vqmA$_{Phage}$* and *qtip*. The endogenous activator of *vqmA$_{Phage}$* transcription that initially launches the phage VP882 lytic process remains unknown. However, given that restoration of all strain 882 QS circuits is not sufficient to launch lysis (S4B Fig), inputs into *vqmA$_{Phage}$* expression in addition to QS must exist.

Beyond lysogenizing strain 882, phage VP882 has been found in two *Salmonella enterica* strains (accession codes: AAFWQT000000000 and AAEKWQ000000000 and [17]), *Shewanella* algae sp. (accession code: LTBI01000119.1), and *V. parahaemolyticus* (accession code: NNHH01000051.1 and [17]). Regarding the additional *V. parahaemolyticus* strain, curiously, host *vqmR-vqmA* is intact, but an 819 bp deletion encompassing *vqmA$_{Phage}$* and *qtip* exists in its phage VP882. As described here, the reverse is the case in strain 882: *vqmA$_{Phage}$* in phage VP882 is intact, but host *vqmR-vqmA* is disrupted. *Salmonella* and *Shewanella* do not possess *vqmA* and *vqmR* genes. Thus, no naturally occurring host that is lysogenic for phage VP882 has been identified that harbors two copies of *vqmA*. We know that both bacterial VqmA and VqmA$_{Phage}$ activate expression of host *vqmR* [8,11], and our work here indicates that VqmR production activates phage VP882 lytic gene expression (albeit modestly and in the presence of a functional LuxO; Fig 4A). Given these two features, we speculate that harboring two copies of *vqmA* is disadvantageous to the host, and bacteria that eliminate one copy of *vqmA*, either from the host or apparently from the phage, receive superior protection from their parasitic viral partner. An alternative possibility is that the phage benefits from this arrangement by limiting superinfection. Specifically, the QS-defective state of the host renders it insensitive to AIs produced by kin bacteria in the vicinal community, curbing QS-mediated induction of the VP882 prophage, and potential costly production of phage particles that lack hosts to infect in a fully lysogenized bacterial community.

Unexpectedly, the Δ91–102 LuxO (i.e., LuxO$_{882}$) variant was present in an additional 87 sequenced *Vibrios*. Indeed, a recent screen for mutations that affect colony sectoring and kin-killing in an environmental isolate of *V. cholerae*, called *V. cholerae* 2740–80, identified a mutation in LuxO with a similar deletion (residues 94–106 and [18]). Analogous to what we report here, this allele drove LCD QS behavior in *V. cholerae* [18], suggesting that in-frame deletions in LuxO confer advantages in different *Vibrios* and in different contexts. Mapping of LuxO$_{882}$ onto the *V. angustum* LuxO crystal structure revealed that the eliminated amino acids in LuxO$_{882}$ reside along interface II between the LuxO catalytic and receiver domains [16]. In the unphosphorylated state, the LuxO catalytic and receiver domains interact via interface II. By contrast, when residue D61 is phosphorylated, interface II is disrupted, fostering LuxO transcriptional activity. Likely, deletion of residues 91–102 permanently disrupts intra-protein domain interactions at interface II, locking LuxO$_{882}$ into its transcriptionally-active state. LuxO mutations, as well as mutations in homologs of OpaR, which also confer the LCD QS state, are common in *Vibrios* in nature. Presumably, these variations in QS system function are driven by selective pressures that deliver fitness benefits under particular conditions [19–21]. One possible example is that mutations that lock *Vibrios* into the LCD QS state prevent the costly production of QS-controlled public goods [21], and here we show protect from phage killing.

Links between host QS and phage-host interactions have been investigated primarily in the context of QS-control of anti-phage defense mechanisms [22–27]. However, a few examples,

including in this present work, demonstrate that temperate phages can garner information from host-produced QS cues to regulate their lysis-lysogeny transitions [8,28,29]. A recent study demonstrated that in *Vibrio anguillarum*, QS represses φH20-like phage p41 lytic development at HCD [30]. By contrast, QS in *V. parahaemolyticus* drives phage VP882 lytic development at HCD, whereas the naturally locked LCD QS state of strain 882 suppresses phage VP882 viral reproduction. Likely, the ramifications of host QS signaling on temperate phage biology are specific to particular phage-host partnerships.

## Methods

### Bacterial strains, reagents, and growth conditions

*E. coli* and *V. cholerae* strains were grown with aeration in Luria-Bertani (LB-Miller, BD-Difco) broth at 37 ˚C. *V. parahaemolyticus* strains were grown with aeration in LB with 3% NaCl at 30 ˚C. Strains used in the study are listed in S2 Table. Unless otherwise noted, antibiotics, were used at: 100 μg mL$^{-1}$ ampicillin (Amp, Sigma), 50 μg mL$^{-1}$ kanamycin (Kan, GoldBio), 50 μg mL$^{-1}$ polymyxin B (Pb), and 5 μg mL$^{-1}$ chloramphenicol (Cm, Sigma). L-arabinose (Sigma) was supplied at final concentrations of 0.02% or 0.2%, as indicated in the figure legends. Ciprofloxacin (Sigma) and DPO were supplied at a final concentrations of 500 ng mL$^{-1}$ and 10 μM, respectively.

### Cloning techniques

All primers and dsDNA (gene blocks) used for plasmid construction and qPCR, listed in S3 Table, were obtained from Integrated DNA Technologies. Gibson assembly with HiFi DNA assembly mix, intramolecular reclosure, and traditional methods were employed for cloning. PCR with iProof was used to generate insert and backbone DNA. Cloning enzymes were obtained from NEB. Plasmids used in this study are listed in S4 Table. Transfer of plasmids into *V. parahaemolyticus* and *V. cholerae* was carried out by conjugation followed by selective plating on LB plates supplemented with appropriate antibiotics.

### Growth, lysis, and reporter assays

Overnight cultures of *V. parahaemolyticus* and *E. coli* were back-diluted 1:1000 and 1:100, respectively, with fresh medium containing appropriate antibiotics prior to being dispensed (200 μL) into 96 well plates (Corning Costar 3904). Arabinose and DPO were added as specified. Wells that did not receive treatment received an equal volume of water. A BioTek Synergy Neo2 Multi-Mode reader was used to measure OD$_{600}$, fluorescence, and bioluminescence. Relative light units (RLU) and relative fluorescence units (RFU) were calculated by dividing the bioluminescence and fluorescence readings, respectively, by the OD$_{600}$ at that time.

### qPCR

Overnight cultures were back-diluted 1:1000 with specified treatments. Subsequently, cultures were grown for an additional 4.5 h prior to 1:10 dilution in water. Samples were heated at 95 ˚C for 20 min to release viral DNA packaged in phage VP882 virions, linear phage VP882, and host genomic DNA from cells. DNA was further diluted 1:100 in water, and 1 μL was used in qPCR reactions. SYBR Green mix (Quanta) and Applied Biosystems QuantStudio 6 Flex Real-Time PCR detection system (Thermo) were used for real-time PCR. Data were analyzed by a comparative CT method in which the *gp69* target gene was normalized to an internal bacterial control gene (*hfq*).

### *in vitro* HALO-cI$_{VP882}$ repressor cleavage and in-gel HALO detection

Assessment of HALO-cI$_{VP882}$ cleavage was carried out in *V. parahaemolyticus* 882 strains as described [8] with modifications. Specifically, overnight cultures carrying the plasmid harboring HALO-cI$_{VP882}$ were diluted 1:200 in medium and grown for 2.5 h with shaking. Cultures were divided in half and administered ciprofloxacin or water as specified. The treated cultures were incubated without shaking for an additional 1.5 h. Cells were collected by centrifugation (16,100 X g for 1 min), resuspended in a lysis buffer containing 1x BugBuster, benzonase, 300 mM NaCl, and 1 μM HALO-TMR (excitation/emission: 555/585 nm). Clarified lysates were loaded onto NEBExpress Ni spin columns (NEB), washed once with lysis buffer containing 10 mM imidazole, and eluted in lysis buffer containing 500 mM imidazole. Eluted samples were subjected to electrophoresis on 4–20% SDS-PAGE stain-free gels. Gels were imaged using an ImageQuant LAS 4000 imager under the Cy3 setting prior to being exposed to UV-light for 7 min and re-imaged under the EtBr setting. Exposure times never exceeded 30 sec.

### Western blot analyses

Western blot analyses were performed as reported [11] with the following modifications: *V. parahaemolyticus* and *V. cholerae* producing C-terminal 3XFLAG-tagged VqmA were back-diluted 1:1000 and harvested at $OD_{600} = 0.2$ and 2.0. *E. coli* producing 3XFLAG-tagged VqmA were back-diluted 1:100 and harvested at $OD_{600} = 2.0$. *V. parahaemolyticus* producing 3XFLAG-tagged LuxO were back-diluted 1:1000 and harvested at $OD_{600} = 2.0$. *E. coli* and *Vibrios* were resuspended in Laemmli sample buffer at final concentrations of 0.006 OD $μL^{-1}$ and 0.02 OD $μL^{-1}$, respectively. Following denaturation for 15 min at 95 ˚C, 5 μL of each sample was subjected to SDS-PAGE gel electrophoresis. RpoA was used as the loading control (Biolegend Inc.). Signals were visualized using an ImageQuant LAS 4000 imager.

### DNA and RNA sequencing

All sequencing was carried out by SeqCenter (formerly Microbial Genome Sequencing Center; MIGS). For preparation of samples for genomic sequencing, overnight cultures of strain 882 were back-diluted 1:1000 and grown to stationary phase prior to harvesting. Genomic DNA was extracted using the DNeasy Blood & Tissue Kit (Qiagen) according to the manufacturer's protocol. For preparation of samples for RNA sequencing, overnight cultures of the 882 strains were back-diluted 1:1000 and grown for 4 h prior to harvesting. Total RNA was isolated using the RNeasy Mini Kit (Qiagen). RNA samples were treated with DNase using the TURBO DNA-free Kit (Thermo).

### Multiple sequence alignment

Genomic DNA sequences of 5534 *Vibrio* strains were downloaded from the GenBank database [31]. A first pre-processing step was performed to discard duplicate genomes and genomes that contained assembly gaps in regions of interest, which yielded 5460 genomes for further analyses. A custom MATLAB (Mathworks, 2022) algorithm was used to automate the sequence-similarity-based search of *vqmR*, *vqmA*, and *luxO* genes in each of the 5460 genomes [32]. The DNA sequence of *vqmR*, and protein sequences of VqmA and LuxO from *V. cholerae* C6706 were used as queries. Multiple sequence alignment of DNA and protein sequences were performed and analyzed in MATLAB and visualized by NCBI Multiple Sequence Alignment Viewer (v1.22.1).

## Phylogenetic analyses

Genomic sequences were annotated using Prokka (v 1.14.6) [33]. A core genome was extracted from Prokka-output GFF files and aligned using built-in MATLAB functions. A post-processing step was next applied to remove regions containing long consecutive gaps or low-quality alignment from the core genome alignment to avoid using false SNPs for phylogeny reconstruction. Pairwise p-distances were calculated based on the reduced core genome alignment and were subsequently used to build a phylogenetic tree using the UPGMA (unweighted pair group method with arithmetic mean) method. Using other distance-based methods, such as WPGMA (weighted pair group method with arithmetic mean), or a maximum parsimony method yielded similar results of phylogeny reconstruction.

## Quantitation and statistical analyses

Software used to collect and analyze data generated in this study consisted of: GraphPad Prism 9 for analysis of growth and reporter-based experiments; Gen5 for collection of growth and reporter-based data; SnapGene v6 for primer design; QuantStudio for qPCR collection; LASX for acquisition of western blots; and FIJI for image analyses. Data are presented as the means ± std. The number of technical and independent biological replicates for each experiment are indicated in the figure legends.

## Supporting information

**S1 Table. Numerical values and associated p-values for heatmaps in Fig 4A.**
(DOCX)

**S2 Table. Strains used in this study.**
(DOCX)

**S3 Table. Primers and gBlocks used in this study.**
(DOCX)

**S4 Table. Plasmids used in this study.**
(DOCX)

**S1 Data. Data points used to make plots that appear in this study.**
(XLSX)

**S2 Data. Raw RNA-sequencing data generated in this study.**
(XLSX)

**S3 Data. Unprocessed gels and blots used in this study.**
(PPTX)

**S1 Fig.** *V. parahaemolyticus* **strain 882 is the only sequenced** *Vibrio* **that lacks** *vqmR* **and harbors** *vqmA,* *vqmA* **expression is not auto-regulated in** *V. parahaemolyticus,* **and** *Vibrio* *vqmA* **promoters cluster into two distinct classes.** (A) Percentage of strains possessing (gray) or lacking (turquoise) *vqmR-vqmA* pairs (*left*), and number of strains analyzed (*right*) for the designated species. Vas: *Vibrio aestuarianus.* Val: *Vibrio alginolyticus.* Van: *Vibrio anguillarum.* Vbr: *Vibrio breoganii.* Vcm: *Vibrio campbellii.* Vch: *Vibrio cholerae.* Vcrl: *Vibrio coralliilyticus.* Vcrr: *Vibrio coralliirubri.* Vcr: *Vibrio crassostreae.* Vcy: *Vibrio cyclitrophicus.* Vdb: *Vibrio diabolicus.* Vdz: *Vibrio diazotrophicus.* Vfl: *Vibrio fluvialis.* Vfr: *Vibrio furnissii.* Vhr: *Vibrio harveyi.* Vjs: *Vibrio jasicida.* Vkn: *Vibrio kanaloae.* Vln: *Vibrio lentus.* Vmd: *Vibrio mediterranei.* Vmtc: *Vibrio metoecus.* Vmts: *Vibrio metschnikovii.* Vmm: *Vibrio mimicus.* Vnv: *Vibrio*

*navarrensis*. Vmg: *Vibrio nigripulchritudo*. Vow: *Vibrio owensii*. Vpch: *Vibrio paracholerae*. Vpr: *Vibrio parahaemolyticus*. Vrt: *Vibrio rotiferianus*. Vsp: *Vibrio splendidus*. Vts: *Vibrio tasmaniensis*. Vvl: *Vibrio vulnificus*. (B) *Left*: Relative P*vqmA_{RIMD}*-*lux* output from *E. coli* carrying arabinose-inducible *vqmA_{882}-3XFLAG*. The treatments - and + VqmA_{882} refer to water and 0.2% arabinose, respectively. RLU as in Fig 2B. *Right*: representative western blot of VqmA_{882}-3XFLAG produced by the *E. coli* in the left panel. RpoA was used as the loading control. (C) Multiple DNA sequence alignment of the intergenic regions between *vqmR* and *vqmA* for the *V. cholerae* clade (Clade 1) and the *V. parahaemolyticus* clade (Clade 2). A representative strain (as designated) was chosen for each species in each clade. Thick gray or red bars indicate, respectively, nucleotides that are identical with or different from the consensus (>50% agreement among aligned sequences). Thin gray lines indicate gaps in the sequence alignments. Scale bar indicates 50 bp. Data in B are represented as means ± std with *n* = 3 biological replicates (*left*) and representative of two independent experiments (*right*).
(TIF)

**S2 Fig. VqmA_{882} binds DPO and promoter DNA.** (A) Protein sequence alignment (ClustalW) showing *V. parahaemolyticus* strain 882 VqmA (VqmA_{882}), *V. cholerae* VqmA (VqmA_{Vc}), and phage VP882 VqmA (VqmA_{Phage}) proteins. Black and gray boxes designate identical and conserved residues, respectively. Numbering indicates amino acid positions. Blue boxes indicate key conserved DPO-binding residues from VqmA_{Vc} (F67, F99, and K101). (B) Relative fold activation of P*vqmR_{RIMD}*-*lux* or P*qtip*-*lux* from Δ*tdh E. coli* harboring arabinose-inducible *vqmA_{882}-3XFLAG*. Fold activation was calculated by dividing the RLU of induced cells (0.02% arabinose and 10 μM DPO) by the RLU of uninduced cells. (C) Relative P*vqmR_{RIMD}*-*lux* and P*vqmR_{Vc}*-*lux* from Δ*tdh E. coli* harboring arabinose-inducible *vqmA_{882}-3XFLAG* (designated VqmA_{882}) or *vqmA_{Vc}-3XFLAG* (designated VqmA_{Vc}), respectively. *E. coli* were treated with either water (black bars) or 10 μM DPO (white bars). All cells were treated with 0.02% arabinose. Data are represented as means ± std with *n* = 3 biological replicates (B, C). RLU as in Fig 2B (B, C).
(TIF)

**S3 Fig. LuxO_{882}-3XFLAG and LuxO^{+}-3XFLAG are functional and 88 *Vibrio* strains possess the Δ91–102 LuxO deletion.** (A) Relative *luxCDABE* output over time from the 882 *luxO_{882}* (cyan), 882 *luxO^{+}* (orange), 882 *luxO_{882}-3XFLAG* (purple), and 882 *luxO^{+}-3XFLAG* (pink) strains. Data are represented as means ± std with *n* = 3 biological replicates. (B) Multiple amino acid sequence alignment of LuxO in *Vibrio* strains that carry the Δ91–102 *luxO* mutation. Gray and red vertical bars indicate, respectively, amino acids that are identical to or different from the consensus (>50% agreement among aligned sequences). White boxes indicate the 91–102 amino acid deletion. Blue vertical lines indicate insertions. Teal indicates *V. parahaemolyticus* strains, green indicates *V. cholerae* strains, and dark blue indicates *V. owensii* strains. Scale bar indicates 50 amino acids (abbreviated AA). All sequences are aligned with respect to the LuxO sequences of *V. parahaemolyticus* RIMD2210633 and *V. cholerae* C6706, which are shown in the first and second row, respectively. (C) As in (B), except the strains carry the Δ67–87 (top) or Δ114–134 (bottom) LuxO mutation.
(TIF)

**S4 Fig. There are no growth defects in the QS-active 882 strains carrying the phage and elimination of *qtip* renders phage VP882 SOS-inducible but not VqmA_{Phage}-inducible.** (A) Growth of the 882 parent (cyan), 882 *vqmR-vqmA_{882}^{+}* (green), 882 *luxO^{+}* (orange), and 882 *vqmR-vqmA_{882}^{+} luxO^{+}* (red) strains. (B) Growth of strain 882 harboring phage VP882 *qtip*::*cm* and arabinose-inducible *vqmA_{Phage}* in medium treated with water (white), arabinose

(gray), or ciprofloxacin (black). Arabinose (0.2%) was used to induce $vqmA_{Phage}$ expression, and ciprofloxacin (500 ng mL$^{-1}$) was used to induce host SOS. Data are represented as means ± std with $n$ = 3 biological replicates (A, B).
(TIF)

## Acknowledgments

We thank all members of the Bassler laboratory for insightful discussions.

## Author Contributions

**Conceptualization:** Olivia P. Duddy, Justin E. Silpe, Chenyi Fei, Bonnie L. Bassler.

**Data curation:** Olivia P. Duddy, Justin E. Silpe, Chenyi Fei, Bonnie L. Bassler.

**Formal analysis:** Olivia P. Duddy, Justin E. Silpe, Chenyi Fei, Bonnie L. Bassler.

**Funding acquisition:** Bonnie L. Bassler.

**Investigation:** Olivia P. Duddy, Justin E. Silpe, Bonnie L. Bassler.

**Methodology:** Olivia P. Duddy, Justin E. Silpe, Chenyi Fei.

**Project administration:** Bonnie L. Bassler.

**Resources:** Olivia P. Duddy, Justin E. Silpe.

**Supervision:** Bonnie L. Bassler.

**Validation:** Olivia P. Duddy, Justin E. Silpe, Chenyi Fei, Bonnie L. Bassler.

**Visualization:** Olivia P. Duddy, Justin E. Silpe.

**Writing – original draft:** Olivia P. Duddy, Justin E. Silpe, Chenyi Fei, Bonnie L. Bassler.

**Writing – review & editing:** Olivia P. Duddy, Justin E. Silpe, Chenyi Fei, Bonnie L. Bassler.

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
