## [Decision Letter · Decision Letter 0]

25 Jun 2023

Dear Dr Bassler,

Thank you very much for submitting your Research Article entitled 'Natural Silencing of Quorum-Sensing Activity Protects Vibrio parahaemolyticus from Lysis by an Autoinducer-Detecting Phage' to PLOS Genetics.

The manuscript was fully evaluated at the editorial level and by independent peer reviewers. All three reviewers appreciated the attention to an important topic but suggested some minor suggestions that we ask you address in a revised manuscript.

Yours sincerely,

Ivan Matic

Academic Editor

PLOS Genetics

Lotte Søgaard-Andersen

Section Editor

PLOS Genetics

Reviewer's Responses to Questions

**Comments to the Authors:**

Reviewer #1: In recent years, there have been exciting new studies (particularly from the Bassler lab) focused on connections between quorum-sensing and symbioses of phage and their prokaryotic hosts. The Bassler lab recently reported a vibriophage VP882, capable of sensing host-produced QS signals that trigger prophage induction. Presumably host QS molecules act as a signal of high prey availability to a prophage. Here, Duddy et al. sequence the genome of the natural host of the VP882 prophage, and find that two host QS systems are disabled. They demonstrate that, if functionality of these systems is restored, virion production and host cell lysis by the phage are enhanced, suggesting that inactivation of host QS confers a fitness benefit to VP882 lysogens.

The data in this paper are beyond reproach and strongly support the conclusions. The paper is also very well-written and was a pleasure to read. I do not have any experimental requests, but have a few points of interest that could be clarified via additional discussion:

1. Is there any correlation between Vibrio isolates that have VqmR-VqmA deletions and those that contain a resident prophage encoding a VqmA homolog?

2. It seems that strain 882 produces DPO at high cell density, as restoration of VqmA-R results in cell density-dependent phenotypes. Is the DPO synthase known? It might be worth pointing out whether it’s present in strain 882.

3. Could the mysterious signal activating transcription of VqmA phage be a direct target of Qrr-mediated translational silencing? Or conversely, is OpaR required for the observed QS-dependent phage phenotype?

4. Once VP882 is established in a host, triggering prophage induction in response to high cell density might be disadvantageous if the producers are lysogenic kin, as these cells would presumably not constitute susceptible prey (since lysogens are typically resistant to superinfection by the phage they already harbor). Therefore it might benefit both the host and phage if the host QS systems are disabled, but phage VqmA remains competent to respond to QS signals from neighboring un-lysogenized cells.

5. Very minor, but it might be worth emphasizing multiple times that prophage VqmA is silent until the unknown signal is produced. I found myself wondering why phage VqmA wasn’t responding to HCD, and this took a while to realize.

Reviewer #2: This is a well conducted and written study characterising loss of function mutations in Vibrio parahaemolyticus that prevent lysis by prophage that detect Quorum Sensing signals. The study is of general interest for understanding how phages regulate their behaviour in response to host ecology, and of more specific interest because the focus is on the natural host of the archetypal “QS eavesdropping” phage that has been extensively studied.

I only have minor comments.

1. End of introduction. Phage could also be beneficial (acting as biowarfare agents), so suggest refrain from use of parasite.

2. Any information linking mutations in vqmA, vqmQ or lux genes and prophage incidence in general across the approx. 5000 genomes?

3. Interesting to carry outs test of associations between presence of defective Lux and VqmA systems across the genomes

4. Page 13, first paragraph. Explain more clearly interactions between Lux and VqmA on phage gene expression. They cant be both additive and epistatic at the same time.

5. The implication throughout is that preventing lysis confers a fitness advantage to the host. But you don’t show this: bacterial growth isn’t affected by QS-induced lysis. Please clarify this pint in the discussion, linking back to point 1.

Reviewer #3: Duddy et al examine the sequence and signalling characteristics of a strain of Vibrio parahaemolyticus of interest because it is the source of a bacteriophage, VP882, encoding VqmAPhage, a protein homolog of the receptor for DPO, a quorum sensing (QS) molecule for Vibrios. In previous work, the Bassler lab first demonstrated that DPO activates a host-encoded protein, VqmA, which leads to group behaviors, including biofilm formation. In work on the phage homolog, they found that VqmAPhage could also bind DPO, and expression of this protein initiated a signalling cascade leading to induction of the phage lysogen. Here, they find that the normal VqmA host signalling is defective in the natural host of VP882, and explore the consequences of this. By “repairing” the mutations, they show that the variants found in this host help to keep the expression of phage genes low, presumably avoiding inappropriate induction of phage lytic growth. Comparative genomics on the components highlighted here provide some novel insights into evolution of the QS pathways among Vibrio strains, although the implications of these have not been fully explored.

1. P. 2, introduction: A rationale provided here for the induction of the VqmAphage pathway and phage lysis at high cell density is that this is a situation where “the probability of infecting the next host cell is maximized.” That would not work well if the other host cells in the vicinity are also lysogens. Given that it is not known what the inducing signal for VqmAphage is, might there be a signal that transmits information about sensitive, non-lysogenic hosts? While this is not a focus of this paper, the work here does suggest that lysogenizing Vibrios that have fully competent QS systems may create problems.

2. Presumably, previously published work on VqmAphage and its ability to activate synthesis of VqmR was not done in this host. It would be useful to spell that out someplace.

3. In reporting the sequence changes in the 882 strain, the focus is on expressionf VqmA882. However, if VqmR is also absent, it would seem that signalling from VqmAphage to VqmR and thus to host behaviors would also be absent in this strain. In swaps into other Vibrios, just the intergenic region was traded, but it was unclear in the text exactly what was “repaired” in some of the experiments. For instance, in Fig. 2b, is it only the vqmA and upstream region that is restored? Is this then different from the strains used later in the paper (Fig. 4).

4. Is the statement “Of the >5000 sequenced strains, strain 882 is the only sequenced Vibrio that lacks vqmR but retains vqmA” referring to the phage version of vqmA? Is it also the only phage VqmA? Maybe it would be worth returning vqmR but not vqmA to the 882 genome, to see the effect of the lysogen in that context.

5. Fig. 2C , 2D, and S1C: These two clades of vqmR/vqmA are interesting. In Clade I, VqmA is expressed constitutively, and a reasonable -10 region seems to be present, although the lack of function for a swap of this sequence into RIMD suggests either something further upstream or possibly another problem. If the transcription start site is known for that, I might be mentioned. In Clade II, no decent conserved -10 is obvious. Is there any evidence where transcription actually starts? Or that transcription (and not translation) is responding to cell density? If the swap to look at what is needed for expression includes the upstream vqmR gene, does this work better? What if it includes first part of the vqmA genes?

6. Fig. 3B, Fig. S3B, Fig. 5B, Fig. 6C: It is not very easy to see what colors are what in many of these graphs (luxO882 and luxOD61A are particularly difficult to tell apart in 3B, for instance). It was also not entirely clear that luxOD61E is full-length luxO, while luxO882D61E is the one with the internal deletion. Maybe spell out more in the legend.

7. Fig. 3F: Since these are all strains with the deletion (other than RIMD2210633), it is difficult to really follow how many times the authors think this deletion arose. It might be more useful to look at trees and close relatives for a few other Vibrios not closely related. If this happened multiple times, aside from some expectation of selection for LCD state, does the DNA sequence around this deletion reveal a likely recombination hot-spot?

8. If LuxO is constitutively active in these strains, locked into an LCD state, is there any loss/change in upstream signalling machinery or downstream output (Qrr sRNAs or OpaR?)? This is an interesting observation but does suggest a very different lifestyle (or maybe a more active VqmA system?).

9. Fig. 4 and epistasis of luxO+. A bit more discussion of why luxO+ is needed for the vqmR/A genes to have any effect would be useful. In Fig. 1A, the implication is that there are parallel but independent inputs into “group behaviors”. Here, the data focuses on phage genes; do group behavior genes show a similar pattern of luxO epistasis? Has that been looked at before in other Vibrios, or does the observation here suggest the evolution of different wiring, maybe due to the LuxO constitutive derivative?

10. Fig. S3B: Again, hard to see color differences for V. cholerae and parahaemolyticus. Since only derivatives with this deletion are shown, other changes are not particularly informative (the species differ). Are there other differences relative to luxO+ strains?

11. Discussion: Is it really true that the same phage was found in Salmonella and Shewanella strains? How clear is it that this is real? Does the phage infect Salmonella? Shewanella? Is it possible these sequences were contaminants?

**Have all data underlying the figures and results presented in the manuscript been provided?**

Reviewer #1: Yes

Reviewer #2: Yes

Reviewer #3: Yes

PLOS authors have the option to publish the peer review history of their article (what does this mean?). If published, this will include your full peer review and any attached files.

Reviewer #1: No

Reviewer #2: No

Reviewer #3: No

---

## [Editor Report · Decision Letter 1]

15 Jul 2023

Dear Dr Bassler,

We are pleased to inform you that your manuscript entitled "Natural Silencing of Quorum-Sensing Activity Protects Vibrio parahaemolyticus from Lysis by an Autoinducer-Detecting Phage" has been editorially accepted for publication in PLOS Genetics. Congratulations!

Yours sincerely,

Ivan Matic

Academic Editor

PLOS Genetics

Lotte Søgaard-Andersen

Section Editor

PLOS Genetics

Comments from the reviewers (if applicable):

**Data Deposition**

http://datadryad.org/submit?journalID=pgenetics&manu=PGENETICS-D-23-00618R1

**Press Queries**

---

## [Editor Report · Acceptance letter]

26 Jul 2023

PGENETICS-D-23-00618R1 

Natural Silencing of Quorum-Sensing Activity Protects Vibrio parahaemolyticus from Lysis by an Autoinducer-Detecting Phage 

Dear Dr Bassler, 

We are pleased to inform you that your manuscript entitled "Natural Silencing of Quorum-Sensing Activity Protects Vibrio parahaemolyticus from Lysis by an Autoinducer-Detecting Phage" has been formally accepted for publication in PLOS Genetics! Your manuscript is now with our production department and you will be notified of the publication date in due course.

With kind regards,

Timea Kemeri-Szekernyes

PLOS Genetics

On behalf of:
